bioengineering/biophysics/biochemistry

fractional viscoelasticity, cell rheology, tissue rheology

**Author for correspondence:**
A. Kabla
e-mail: ajk61@cam.ac.uk

# A unified rheological model for cells and cellularised materials

A. Bonfanti[1], J. Fouchard[2], N. Khalilgharibi[2,3],
G. Charras[2,4,5] and A. Kabla[1]

[1]Engineering Department, Cambridge University, Cambridge, UK
[2]London Centre for Nanotechnology, [3]The Centre for Computation, Mathematics and Physics in the Life Sciences and Experimental Biology (CoMPLEX), [4]Institute for the Physics of Living Systems, and [5]Department of Cell and Developmental Biology, University College London, London, UK

 AB, 0000-0003-2185-4913; NK, 0000-0003-4598-5838;
AK, 0000-0002-0280-3531

The mechanical response of single cells and tissues exhibits a broad distribution of time-scales that often gives rise to a distinctive power-law rheology. Such complex behaviour cannot be easily captured by traditional rheological approaches, making material characterisation and predictive modelling very challenging. Here, we present a novel model combining conventional viscoelastic elements with fractional calculus that successfully captures the macroscopic relaxation response of epithelial monolayers. The parameters extracted from the fitting of the relaxation modulus allow prediction of the response of the same material to slow stretch and creep, indicating that the model captured intrinsic material properties. Two characteristic times, derived from the model parameters, delimit different regimes in the materials response. We compared the response of tissues with the behaviour of single cells as well as intra and extra-cellular components, and linked the power-law behaviour of the epithelium to the dynamics of the cell cortex. Such a unified model for the mechanical response of biological materials provides a novel and robust mathematical approach to consistently analyse experimental data and uncover similarities and differences in reported behaviour across experimental methods and research groups. It also sets the foundations for more accurate computational models of tissue mechanics.

## 1. Introduction

As part of their physiological function, single cells and tissues are continuously exposed to mechanical stress. For example, leukocytes circulating in the blood must squeeze through small capillaries, and the epidermis must deform in response to movements of our limbs.

During development, mechanical forces initiate morphogenetic processes involving epithelial growth, elongation or bending, acting as cues to coordinate morphogenetic events [1]. Epithelial cell sheets are also continuously subjected to deformation as part of normal physiology. For instance, lung epithelial cells are exposed to fast cyclical mechanical stress during respiration [2], while epithelia lining the intestinal wall or those in the skin can experience long lasting strain [3]. Failure to withstand physiological forces results in fracture of monolayers which may lead to severe clinical conditions, such as hemorrhage or pressure ulcers [3–6]. Despite significant progress with the experimental characterization of cell and tissue mechanics, understanding the role of mechanical forces in development and pathology is hampered by the lack of a unified quantitative approach to capture, compare and predict the complex mechanical behaviours of tissues, cells, and sub-cellular components across all physiologically relevant time-scales. Such a framework would also enable us to assess the effects of pharmacological treatments on tissue mechanical response without necessitating experimental characterization of the tissue response to all loading conditions, something important for tissue engineering and the design of palliative treatment strategies.

In recent years, experimental characterization of the mechanical behaviour of single cells and tissues has revealed a complex set of mechanical behaviours in response to deformation [7–10]. For example, both single cells and tissues often display multiphasic responses in stress relaxation and creep tests, which comprise a combination of power-law and a faster relaxation regime modelled as an exponential behaviour. Power-law responses are commonly observed in biomaterials and are thought to originate from their complex hierarchical structure [11–14]. These behaviours cannot be easily modelled using traditional linear viscoelasticity, where constitutive rheological models result from combinations of elastic springs and viscous dashpots that translate into sets of linear ordinary differential equations [9,15–18]. In this framework, power-laws can only be implemented through a large numbers of linear elements [19], making this approach impractical and uninformative. Empirical functions have been introduced to overcome this challenge [12,20,21], but the lack of underlying material model, in the form of a computable differential equation, prevents the direct comparison of data collected under different loading conditions.

One potential approach for modelling the mechanics of materials presenting power-law behaviours is fractional calculus [22]. This relies on the introduction of a mechanical viscoelastic element called a spring-pot whose behaviour is intermediate between a spring and a dashpot [23]. This element based on fractional derivatives captures, with only two parameters, the broad distribution of characteristic times [24] typical of the mechanical response of cellularised materials. This element has recently been combined with traditional elements to model more complex rheological behaviours, referred to as generalized viscoelastic models [25].

In this paper, we examine the potential of generalized viscoelastic models for modelling biological materials by combining traditional rheological elements with the spring-pot. With only four parameters, we capture the time-dependent response of single cells and epithelial monolayers. Using parameters extracted from relaxation tests, we are able to predict the response of the same material to creep and ramp deformations with no further fitting, and relate the model parameters to single cell characteristics as well as recent measurements of cortical rheology.

## 2. A constitutive model for epithelial monolayers

One widely used model system for studying tissue mechanics is the epithelium monolayer. We focused on Madin-Darby Canine Kidney (MDCK) cell monolayers devoid of substrate, of typical width of 2 mm and suspended between two rods at a distance of 1.5 mm. Despite the absence of a substrate, cells still retain epithelial characteristics [9]. Such a material has now been extensively studied [26,27], and the effect of pharmacological treatments on rheological properties characterized [21]. The advantage of such a simplified system lies in the fact that the tension is transmitted only through the intercellular junctions and the cytoskeleton, but not the extracellular matrix.

The relaxation response (stress response to a step in strain, see the electronic supplementary material, S1) consists of an initial power-law phase in the first ~5s, followed by an exponential phase that reaches a plateau at 60s [21] (see figure 1). By using simple visco-elastic model such as the standard linear solid model (inset in the electronic supplementary material, figure S5), it is possible to capture the time-scale at which the plateau is reached [9]. However, the power-law behaviour at short time-scale [21] is not properly accounted for by standard viscoelastic models (electronic supplementary material, figure S5 top row). The qualitative analysis of the relaxation response highlights the relevant regimes and parameters needed to describe the material's behaviour: (i) the level of the final plateau in figure 1a, (ii) the time-scale beyond which the relaxation function becomes negligible (exponential

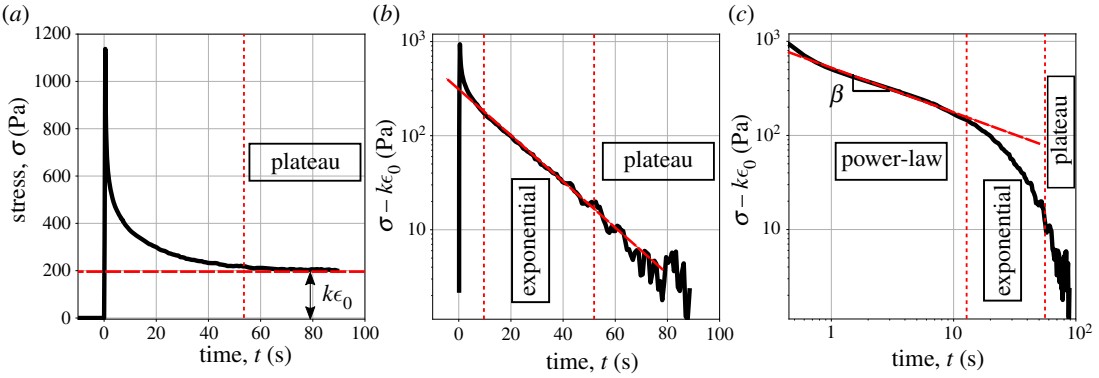

**Figure 1.** Representative experimental data of the stress relaxation of epithelial monolayers depleted of substrate (previously reported in [21]). (*a*) An example of relaxation curve which highlights the final plateau. After removing the plateau, the relaxation curve is plotted in semilogarithmic scale (*b*) and logarithmic scale (*c*) to identify respectively the exponential and power-law behaviours.

cut-off, slope in figure 1*b*), (iii) the power-law exponent at short time scales, and (iv) the overall magnitude of the relaxation response (slope and intercept in figure 1*c*). This qualitative analysis will now inform the development of a novel rheological model tailored to capture these four independent components of the response.

A branch of mathematics called fractional calculus provides conceptual and numerical tools well suited to capture power-law behaviours [28,29]. In traditional calculus, a function can be differentiated $n$ times, where $n$ is an integer. For viscous (fluid-like) materials, the stress is proportional to the first time derivative of the strain, where $n = 1$. For elastic (solid-like) materials, the stress is proportional to the strain, which can be seen as the zero-th time derivative of the strain $n = 0$. Fractional calculus generalizes the differentiation process such that the number $n$ can now be real (see the electronic supplementary material, S2). With the spring-pot fractional element, the stress is proportional to the $\beta$ derivative of the strain, where $0 \leq \beta \leq 1$:

$$\sigma(t) = c_\beta \frac{d^\beta \epsilon(t)}{dt^\beta}, \tag{2.1}$$

where $c_\beta$ is a constant dependent on the material and $d^\beta/dt^\beta$ is the fractional derivative operator. When $\beta = 0$, the material behaves like a spring, and, when $\beta = 1$, like a dash-pot. As $\beta$ varies from 0 to 1, the response of the material continuously transitions from elastic to viscous behaviour and if a step change in stress or strain is applied, the response exhibits a power-law. Mathematically, the response is only defined by an integral over time, leading to strong history dependence, referred to as the hereditary phenomena (electronic supplementary material, S2). Despite this complexity, the spring-pot still lies in the linear viscoelastic framework, enabling us to greatly simplify the analysis of the data and make predictions. For dimensional consistency, the unit of the constant $c_\beta$ is (Pa s$^{-\beta}$), and therefore it does not have a straightforward physical meaning, although, it has been argued that it may represent a measurement of the firmness of the material [30].

The spring-pot can be combined with other rheological elements to generate a rich set of behaviours [31]. Configurations explored so far were mostly selected for their mathematical simplicity, rather than relevance to particular physical systems [25,32]. Here, we adopt a phenomenological approach based on our qualitative description of the material's behaviour, aiming to capture both its short and long time-scale response. At long time-scale, the stress response shows a plateau (figure 1*a*). Hence the model requires a spring in parallel with a dissipative branch that would not carry any tension in steady state. As shown in the electronic supplementary material, figure S5, the standard linear solid model successfully captures the long time-scale response while it omits the power-law response at short time-scale. The element that reacts immediately after application of the strain is the spring in series to the dashpot. Therefore, to capture the presence of a power-law response at short time-scale, such spring is replaced with a spring-pot. The fractional model introduced in the dissipative branch is a special case of a known combination referred to as a fractional Maxwell model (FMM) [31]. Note that the FMM model is known to behave asymptotically as a product between a power and an exponential function [33], as observed experimentally for the epithelial monolayers. The constitutive equation for the fractional material model introduced here in figure 2*a* is reported in the electronic

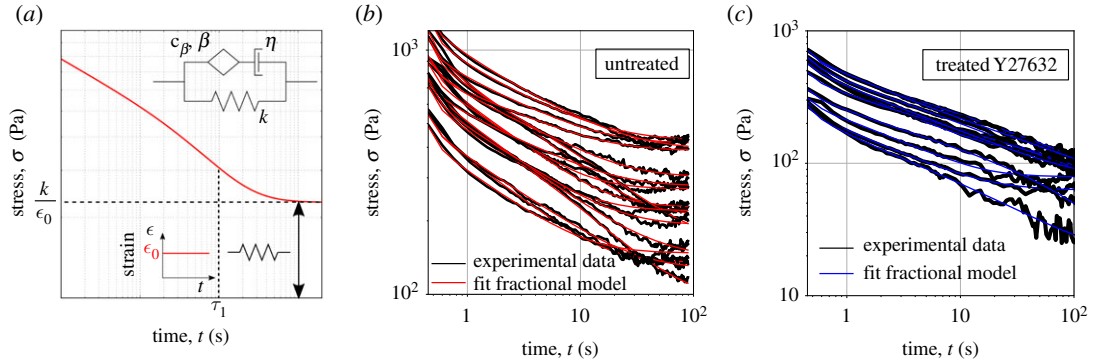

**Figure 2.** Fractional viscoelastic model for epithelial monolayers: constitutive model and stress relaxation behaviour. (a) Diagrammatic representation of the fractional rheological model and qualitative behaviour of its stress relaxation modulus. The three-element fractional model is fitted to the relaxation data for (b) untreated epithelial monolayers (black curves are the experimental data, while the red curves represent the fit) and (c) monolayers treated with an inhibitor of contractility, Y27632 (the black curves are the experimental data, while the blue curves are the fits). Note that the monolayers are loaded with the constant rate of strain $\dot{\epsilon} = 75\% \, s^{-1}$ and the time is set to zero at the beginnings of the loading ramp.

supplementary material, S2. In what follows, such a qualitative approach is validated against experimental data, leading to a predictive model of the material's behaviour.

# 3. The generalized fractional viscoelastic model characterizes the biphasic stress relaxation response

The relaxation modulus (stress response to a unit strain of deformation) of the fractional network model introduced above can be derived analytically. Because the relaxation modulus of two elements in parallel is given by the sum of their relaxation moduli, the relaxation modulus $G(t)$ of the novel viscoelastic model presented in figure 2a is obtained by adding the relaxation modulus of the FMM to the stiffness of the spring, which results in

$$G(t) = c_\beta t^{-\beta} E_{1-\beta,1-\beta}\left(-\frac{c_\beta}{\eta}t^{1-\beta}\right) + k, \tag{3.1}$$

where $E_{a,b}(z)$ is the Mittag-Leffler function, a special function that arises from the solution of fractional differential equations (see the electronic supplementary material, S2). The qualitative behaviour of the relaxation modulus is plotted in log-log scale in figure 2a. Because the argument of the Mittag-Leffler function is non-dimensional, we can identify a characteristic time $\tau_1$, given by

$$\tau_1 = \left(\frac{\eta}{c_\beta}\right)^{1/(1-\beta)}, \tag{3.2}$$

which approximates the time at which the dashpot comes into play, accelerating the convergence of the relaxation modulus towards the plateau value.

To assess the validity of the fractional model, we used it to fit the relaxation response of epithelial monolayers. In agreement with the qualitative analysis of the curves (figure 1), the four parameters involved in equation (2.1) account for the experimental data (see figure 2b), successfully capturing all time-scales. To further test the model and explore how model parameters relate to key subcellular structures, we investigated the relaxation response of epithelial sheets after pharmacological treatments to affect contractility and actin polymerisation, as recently presented by Khalilgharibi et al. [21]. The model could successfully capture the rheological response of treated monolayers, allowing us to quantify the material's behaviour in a systematic manner. Affecting actin polymerisation with CK666 (preventing polymerisation through Arp2/3) and SMIFH2 (preventing formin based actin polymerisation) has no a strong impact on the model parameters (see more details in the electronic supplementary material, S4). However, reducing contractility with the Y27632 ROCK inhibitor has a significant effect on model parameters (figure 2c), when compared with the corresponding control case (dimethyl sulfoxide treated; electronic supplementary material, figure S7). The viscosity $\eta$ of Y27632 treated monolayers doubles, in line with cell-scale findings suggesting that dynamic

contraction of actin filaments increases cell fluidity [8]. By contrast, a reduction of the stiffness $k$ was observed (electronic supplementary material, figure S7(d)), which suggests that acto-myosin contractility mainly plays a role in stress dissipation at long time-scale, consistent with the conclusions previously presented by [8,21]. We also examined the role of two most abundant actin crosslinkers (filamin A and $\alpha$-actinin 4). In agreement with previously reported results [21], no significant variations in the relaxation response of monolayers is observed (see statistical analysis of the fitted parameters in figures S7 and S8 in the electronic supplementary material). Overall, this analysis demonstrates that the fractional rheological model introduced in this section is well suited to study the stress relaxation of MDCK monolayers. It is in particular possible to collapse all data into a master curve that summarises the generic response of both treated and untreated monolayers (see the electronic supplementary material, S5).

# 4. The generalised fractional viscoelastic model predicts the response to different loading conditions with no further fitting

The model relies on the assumption that the material behaves linearly. To identify the linear domain, we examined the stress response at different strain amplitudes ranging from 20% to 50%. We can observe that the material parameters are almost constant until roughly 30% (see the electronic supplementary material, figure S10), which provides an upper bound of the linear domain where we expect the model to be valid. Within this range, we can assess the predictive power of our rheological description of epithelial monolayers. We extracted a distribution of parameters from the stress relaxation data and used them to estimate the response of the material to different forms of mechanical stimulation. Good agreement between predictions and experiments over a broad range of testing protocols would signify that our description represents a constitutive model whose parameters can be seen as material properties. We first consider the stress response to a strain ramp applied at constant strain rate ($1\% \, \text{s}^{-1}$). The predicted response for the untreated monolayers is shown in figure 3a, with 95% confidence interval (see the electronic supplementary material, S3 and S6 for details about prediction and statistical analysis, and see figure S6(a) for Y27632 treated monolayers). The experimental results and the model predictions are in agreement with no free parameters.

Similarly, we can challenge the model by predicting and validating the deformation response $J(t)$ of the epithelial monolayers to a unit step in stress, a test usually referred to as a creep experiment (see the electronic supplementary material, S3). For linear viscoelastic materials the relation between relaxation $\tilde{G}(s)$ and creep $\tilde{J}(s)$ moduli in the Laplace domain is relatively simple, and given by $\tilde{G}(s)\tilde{J}(s) = s^{-2}$. After transforming the relaxation modulus in equation (3.1) in the Laplace domain, we find:

$$\tilde{J}(s) = \frac{1}{ks} \frac{1 + (\tau_1 s)^{1-\beta}}{(\eta/k)s + 1 + (\tau_1 s)^{1-\beta}}. \tag{4.1}$$

To obtain the solution in the time domain $J(t)$, the inverse Laplace transform of the equation above is performed numerically.

The creep response is richer than the relaxation response, for which $k$ only added a simple offset to the stress. Here, because the imposed load can continuously redistribute between the two branches of the model, $k$ is involved in the dynamics. We can indeed identify in the creep response an additional time-scale $\tau_2$ involved in the response: $\tau_2 = (\eta/k)$. We therefore have one additional dimensionless parameter which controls the shape of the creep response $\xi = \tau_1/\tau_2$. The value of $\xi$ leads to qualitatively different responses as plotted in figure 3b. (i) If $\xi < 1$, at short times we first observe a power-law behaviour arising from the spring-pot followed by an exponential regime where the dashpot dominates. The transition from spring-pot-dominated to dashpot-dominated regime is governed by the characteristic time $\tau_1$, as for the relaxation response. While the deformation increases, the spring eventually becomes relevant and the system tends towards the plateau as a Kelvin-Voigt model with a characteristic time $\tau_2$. (ii) If $\xi > 1$, the spring saturates before the transition to the dashpot occurs in the dissipative branch. Hence, the model behaves as a Fractional Kelvin-Voigt model with a characteristic time $\tau'_2 = (c_\beta/k)^{1/\beta}$, which can be expressed as $\tau'_2 = \tau_1^{(\beta-1)/\beta} \cdot \tau_2^{1/\beta}$. $\tau_1$ is irrelevant in this regime. (iii) If $\xi \approx 1$ the transition from the spring-pot to the dashpot corresponds to the time at which the spring becomes relevant. Therefore, the transition from the spring-pot to the Kelvin-Voigt model occurs with characteristic time $\tau_1 \approx \tau_2$.

The model is now used to predict the response of monolayers when subjected to a step in stress using the material parameters derived from the relaxation experiments. We performed new experiments to test

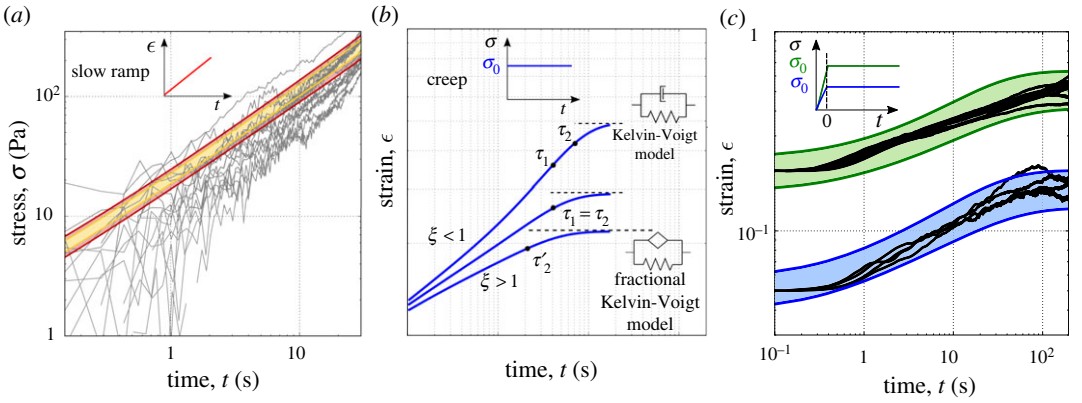

**Figure 3.** Prediction of epithelial monolayers response to different mechanical stimuli using the mechanical parametrization determined from stress relaxation experiments with no further fitting. (*a*) Predicted stress response of the untreated monolayers when subjected to a slow stretch (1% s$^{-1}$). The predicted responses (95% confidence interval red areas and 70% yellow areas) are in agreement with the experimental data (black curves). The upper and lower limits of the predicted response are obtained by considering the standard error for each mechanical parameter. (*b*) Sketch of the creep compliance of the generalized fractional viscoelastic model. Three possible qualitative behaviours can arise dependent on the relative values of the two characteristic times. (*c*) Creep response of the untreated epithelial monolayers. Two loadings are tested, 170 Pa (blue area) and 470 Pa (green area). These loads correspond to an initial strain respectively of 5% and 20%; therefore, the linearity assumption still holds. Note that the initial response of the creep is different from (*b*). This is owing to the ramp during the initial phase (see the electronic supplementary material, S1).

our model's predictive power (see electronic supplementary material, S1). In these experiments, we subjected monolayers to a stress which was maintained constant after an initial short ramp in strain (see inset figure 3*c*). Strikingly, the experimental data falls well within the 95% confidence interval of the predicted response with no free parameters (details in the electronic supplementary material, S3 amd S6) (figure 3*c*).

MDCK monolayers seem to exhibit very different creep behaviour when myosin II activity is reduced. Whilst untreated monolayers reach a steady strain value after about 100 s, Y-27632 treated tissues continue to flow in a power-law manner (see electronic supplementary material, figure S6), suggesting a qualitative difference between the two systems. This apparent contrast is however properly accounted for by the model. The predicted creep responses, calculated using the parameters obtained from figure 2, are in good agreement with the experimental data with no free parameters. Based on the parameters obtained from relaxation experiments on Y-27632 treated monolayers, $\tau_1$ and $\tau_2$ are both much larger than in the untreated case, and now comparable in value with the duration of the measurement (see the electronic supplementary material, table S2). Furthermore, the reduction of $\xi$ in the treated case, down to 0.7 compared to about 1 in the untreated case (electronic supplementary material, figure S6(c)), would change the shape of the creep curve and give the impression that creep accelerates rather than saturates at times close to $\tau_1$ (figure 3*b*). This analysis illustrates how modelling can bring consistency across systems that may at first appear qualitatively different, in particular when observed over a finite experimental time.

# 5. Usage of the model beyond epithelial monolayers

Building on the work on MDCK monolayers, we may now consistently analyse data across biological systems, and pull information from different research groups, working with different mechanical testing protocols. For instance, we can show that the model successfully captures the relaxation response of single isolated cells, such as epithelial MDCK cells (figure 4*a* [21], details on experimental setup in the electronic supplementary material, S7) and articular chondrocytes (figure 4*b*, original data presented by Darling *et al.* [34]).

Looking at sub-cellular components, the main factors controlling the mechanical properties of cells and tissues have been characterized experimentally already. Fischer-Friedrich *et al.* [8] have recently performed oscillatory compressions of HeLa Kyoto cells during mitosis by using an atomic force microscope cantilever and analysed the data to extract the rheological behaviour of the cell's cortical actin network. To allow the direct comparison between our constitutive model and the rheological data introduced by Fischer-Friedrich *et al.* [8], we calculated the analytical expression of the complex

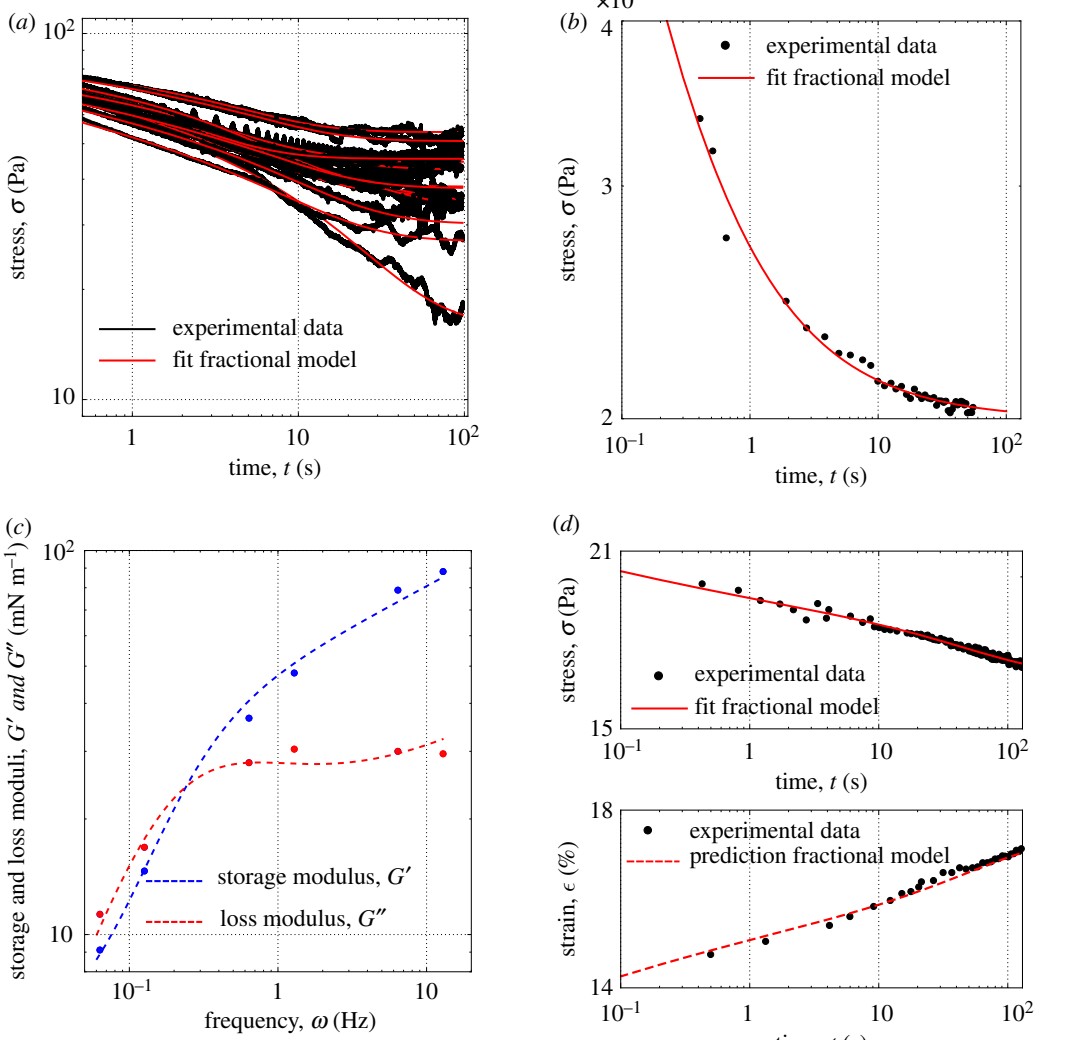

**Figure 4.** The unified model captures the mechanical behaviour regardless of the experimental set-up, the type of material and the length-scale. (a) Fitting of the relaxation response of single epithelial cells presented by Khalilgharibi *et al.* [21]. Each line is a different cell. (b) Fitting of the relaxation response of articular zone chondrocytes (original data from [34]). (c) Fitting of the storage and loss modulus of HeLa Kyoto cells (original data from [8]) using the model presented here. The blue and red dashed lines are respectively the fitted storage and loss modulus while the dots are the experimental data. (d) Fitting of the relaxation response of collagen fibrils (top graph) and prediction of the creep response (bottom graph). Original data from [35]. Fitted parameters in the electronic supplementary material, table S2.

modulus $G'(\omega) + iG''(\omega)$ associated with our model (see the electronic supplementary material, S8) and fitted the experimental data again with a good agreement as shown in figure 4c.

Likewise, with the modelling framework presented here we have been able to capture the relaxation response of collagen fibrils (figure 4d top). Furthermore, using the parameters extracted from fitting the relaxation data, we have been able to predict their creep behaviour (figure 4d bottom); a quantitative link that was absent in the original paper presented by Shen *et al.* [35].

As many biomaterials exhibit power-law rheology, examples where the generalized fractional viscoelastic model successfully captures their behaviour abound in literature—e.g. blastomere cytoplasm and yolk cell rheology (see the electronic supplementary material, figure S13(c,d)). Studies have also reported simpler qualitative creep and relaxation behaviours for single cells, such as a single power-law or a two-power-law behaviours [36]. These responses are embodied as special cases of the presented generalized model (negligible stiffness and/or large viscosity). To illustrate this, we fitted the power-law response of single immune cells (electronic supplementary material, figure S14). We could also capture the creep response of a single muscle cell exhibiting a power law response whose slope increases at long time-scale. The model renders this behaviour through a transition from a spring-pot dominated response

at short time-scales to a viscous behaviour at long time-scale. Storage and loss moduli could then be predicted for this system (electronic supplementary material, figure S13 (a,b)). This sets the fractional viscoelastic model presented here as a promising tool to support a unified description of the mechanical response of a broad variety of biological tissues.

# 6. Links with biophysical analysis

Fractional models capture with a few parameters complex power law behaviours commonly associated with a broad distribution of relaxation times. A single spring-pot element is for instance sufficient to model a power law response, with two parameters. The complexity and richness of this element is apparent in the way the fractional derivatives are calculated; whereas the response of traditional elements (spring and dashpot) only depends on the instantaneous evolution of the strain, a hereditary time integral (electronic supplementary material, equation S1) is required to evaluate fractional derivatives, accounting for the strong history dependent effects associated with power law rheology.

Combining the spring-pot element with other rheological components allows us to capture the different regimes observed experimentally. Based on dimensional analysis, we identified in particular two characteristic times and an effective stiffness involved in the relaxation and creep functions. These physical quantities provide convenient handles to compare our results with other physical studies of the rheology of cells and tissues. For example, Fischer-Friedrich *et al.* [8], after observing that a power law function failed to capture the dynamic response of HeLa Kyoto cells, considered a minimal rheological model based on a flat distribution of relaxation times up to a certain cut-off time. The relaxation spectrum of the fractional model introduced here, using the parameters as obtained from the fitting of the complex modulus in figure 4c reproduces this phenomenology ($\tau_1$ being the cut-off time) and is in good agreement with the experimental data (electronic supplementary material, figure S15). The fractional model captures in particular the presence of long time-scales in the spectrum that play an important role in the creep response. The re-analysis of [8] enables us to compare two different biological systems, Hela and MDCK, and identify consistent behaviours despite the use of distinct material characterisation approaches. Parameter values for each single cell type are of the same order of magnitudes, and in both cases, a reduction of myosin activity leads to an increase of the characteristic or cut-off time-scale.

The choice of rheological model is also likely to influence our interpretation of experimental observations. Bonakdar *et al.* [20] performed local measurements of the rheology of mouse embryonic fibroblasts by imposing cycles of loading at constant force followed by relaxation without force applied, using a fibronectin coated magnetic particle linked to the cytoskeleton through the cell membrane (figure 5a). They observed a power-law response during both loading and relaxation, but showed that these two phases could not be accounted for by the same rheological behaviour (i.e. by a unique spring-pot element); the recovery amplitude was too small compared to the loading amplitude. They interpreted this fact as evidence for a novel plastic signature that would itself decrease in magnitude with the strain history of the material. Their proposed rheological model is by definition non linear and therefore qualitatively very different from the picture painted above for MDCK and Hela cells. We nonetheless tried to adjust our model to their data and looked for aspects of the response we could successfully capture. As it is difficult to scale the forces and displacements of the probe into stress and strain inside the cell, we normalised the displacements and focused on reproducing the overall shape and time-scales of the curve.

At first, we only considered a simple power-law behaviour (i.e. a single spring-pot) with an exponent corresponding to the mean value extracted from MDCK cells; we confirmed the observations of Bonakdar *et al.* that a power-law alone cannot reproduce the progressive drift of the bead displacement observed experimentally (figure 5b). Introducing a simple viscous element in series with the spring-pot would produce a behaviour that is more consistent with experimental data. Using the values of the spring-pot firmness $c_\beta$ and dashpot viscosity $\eta$ that we measured for MDCK single cells, the first couple of cycles seem well reproduced, but the displacement then drifts linearly over time and rapidly exceeds the experimental trend (figure 5c), as expected from the behaviour of a dashpot. Our proposed fractional model for single cells and tissues includes a spring in parallel with the branch previously considered, which would intuitively slow down and eventually stop the drift. We therefore tested the complete fractional model with MDCK single cell parameters (figure 5d). The spring value appears to be too large to allow significant drift to happen. We found however that an intermediate spring value corresponding to about 10% of the MDCK single cell parameter provides a

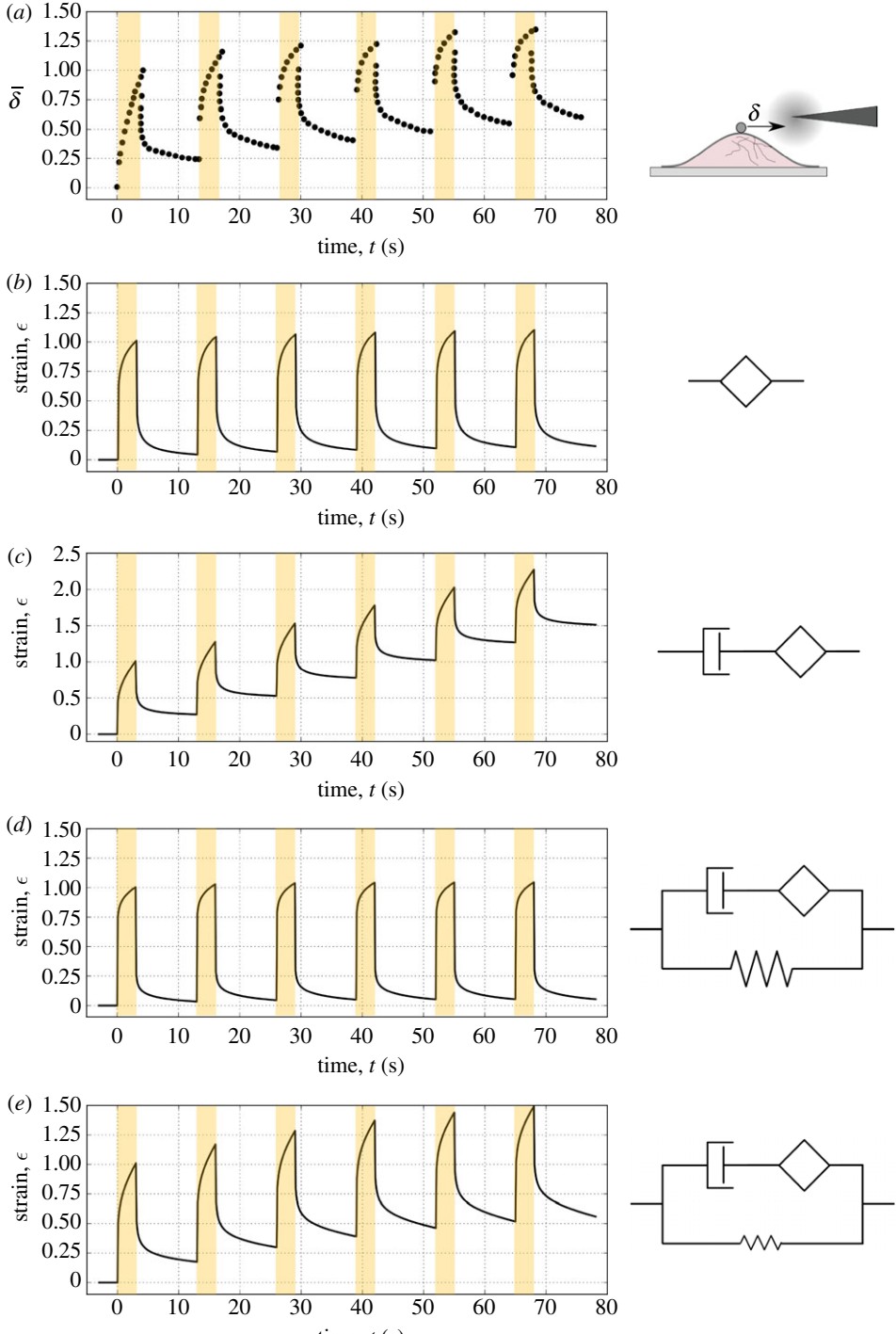

**Figure 5.** Analysis of local rheological data. (*a*) Incomplete recovery of a single cell deformation to alternating force cycles. Digitized data from Bonakdar *et al.* [20]. A magnetic bead is attached to the cytoskeleton via integrin-type adhesion receptors through which a force parallel to the cell is applied (force on in dark areas). Note that the displacement is normalized with respect to the first peak value because it is difficult to identify the interaction of the bead with the cell and therefore convert the displacement into strain. By using the material parameters of MDCK single cells (electronic supplementary material, table S2), we predict the response to cycles of loading using the fractional models and we compare the qualitative responses normalized with respect to the first peak (*b*–*e*) with the experimental data (*a*). Prediction of the response to cyclic loading with the spring-pot only (*b*), with a dashpot in series to the springpot (*c*) with the novel model, (*d*) and with a reduced amount of long time-scale elasticity ($k = 20$ Pa, 10% of the fitted value of $k$) (*e*).

good agreement with Bonakdar's data. In this model, although the dashpot clearly represents a dissipative aspect of the rheology, we did not need to invoke non-linear behaviours and plasticity, but simply adjust a linear model already validated on different cell types. Although this analysis may not

fully account for the non-linear plastic behaviour observed in mouse embryonic fibroblasts, it illustrates that a number of qualitative distinctive traits, reported in different systems and under different experimental conditions, may be in fact largely consistent with each other. In what follows, we will attempt to shed light on the physical meaning of the spring constant and interpret its low value in Bonakdar's micro-rheology experiments.

A unified language to consistently capture the response of materials over a broad range of time-scales and systems helps identify the physical and biological significance of rheological behaviours, one of the current key challenges in mechanobiology [37]. For instance, by comparing the relaxation response of single MDCK cells with MDCK monolayers, we noticed that they display similar behaviours (figure 4$a$). However, looking at the parameter values (see the electronic supplementary material, table S2) reveals that monolayers exhibit a higher stiffness $k$, firmness $c_\beta$, and viscosity $\eta$. Quantifying these mechanical properties raises novel questions, and we can only speculate at this stage about the reasons. The origin of the higher firmness of the spring-pot and viscosity $\eta$ may for instance be related to changes in the internal cell organization when cells form intercellular junctions to integrate into a monolayer.

The stiffness $k$ is the parameter on which we currently have the most insight. Cortical contractility, i.e. the generation of active stresses in the cell cortex owing to ATP dependent myosin activity, has been already associated with increase in elastic modulus in monolayers and single cells [21,27]. In monolayers, contractility also controls the emergence of a macroscopic pre-stress. Such contribution is not directly accounted for here, but could be integrated in the model as demonstrated in [27]. Mechanistically, stiffness emerges at the scale of a single cell owing to the overall increase of the cell cortex area when the cell is squashed or stretched [8]; the stronger the cortical tension is, the more work is required to extend the cortex area, similarly to a soap bubble or froth exhibiting solid properties despite the fluid nature of its components. This explains why Y27632 treatment has such a strong impact on the stiffness $k$ in both single cells and monolayers. However, given the local nature of Bonakdar *et al.*'s measurements, we would not expect that the probe displacement would cause a significant overall increase in the cell cortex area; the elastic term $k$ is therefore likely to be irrelevant as a first approximation, as observed in figure 5. We can only speculate what may control the residual $k$ value in this case. It has been shown previously that during application of a local stress through fibronectin coated beads, cells respond with a local strengthening of the cytoskeleton linkages [38].

The analysis above is an example of a top-down approach to capture and understand tissues and cell mechanics. By extracting and comparing parameters that uniquely characterize the material for different systems and considering a range of pharmacological treatments, we could highlight generic behaviours and key features affected by specific biochemical processes. This helps shape hypotheses and interpretations of these behaviours. A deep understanding would nonetheless require linking these approaches with bottom-up physical models that can explain how observed behaviours would emerge from underlying micro-structural processes [39,40]. A number of theoretical approaches exists, including the soft glassy rheology initially used by Fabry *et al.* [12] to relate the power-exponent to the fluidity of cells, and the glassy worm like chain models which consider the network dynamics of the cytoskeleton (interaction between links, and breaking and reformation of links) to explain the fluidisation behaviour typical of biomaterials. The ultimate aim would be to bridge the top-down approach presented here with physical models to understand the underlying processes giving rise to the rich qualitative behaviour (regimes presented in figure 1) observed in tissues and cell mechanics.

# 7. Conclusion

We have presented a phenomenological model for epithelial monolayers that captures the biphasic nature of their stress relaxation dynamics over their full physiological functional range. From a qualitative analysis of the monolayers' relaxation response, we combined traditional viscoelastic elements with the fractional spring-pot to propose a novel linear model suitable to fit experimental data up to 30% of deformation. The model, calibrated against experimental relaxation data of suspended MDCK monolayers, could predict the monolayer's response to other mechanical tests with good accuracy. This confirms that our model provides a constitutive description of the material and that the fitting parameters are proper material properties independent of the type of deformation or force applied to the material. We further demonstrated the model's suitability to analyse and compare rheological behaviours across many systems and length scales, within the same unifying framework. It allows us to ask more direct questions regarding the biological significance of the parameters involved, and

opens the door to more systematic theoretical analysis, capturing both power-law regimes and intrinsic time-scales in the response.

Beyond enhancing our understanding of biological systems, a unified rheological model for biomaterials is also crucial in the medical and engineering fields. Correlating changes in the mechanical response of tissues to their biological state has been long considered as a promising marker-free method for cancer diagnosis [41–43]. In the context of regenerative medicine, rheological phenotypes provide a suitable metric to assess the similarity between tissue engineered constructs produced *in vitro* and their natural counterparts. As the parameters of the model are material properties that are independent of the method of measurement, they are promising mechanical signatures of the tissue and its condition to be used for diagnosis or as a target for regenerative medicine.

A practical limiting factor for the widespread application of the model presented here is the mathematical complexity of fractional derivatives, and the current lack of user-friendly numerical methods to perform such analysis without expertise in fractional calculus. Such tools have been recently released in the public domain by our group [44]. This will ensure a broader adoption of fractional models for a rapid and systematic analysis of experimental data, as well as future integration within numerical packages and finite element software.

Data accessibility. The new datasets supporting this article have been uploaded as part of the electronic supplementary material in the Dryad Digital Repository: https://doi.org/10.5061/dryad.s853qg7 [45].

Authors' contributions. A.B and A.K. designed the rheological model. N.K. carried out the relaxation and ramp experiments. J.F. and G.C designed the creep experiments. J.F. carried out the creep experiments. A.B and A.K. performed data analysis. J.F. and G.C. contributed to physical interpretation of the data. All authors discussed the results and manuscript.

Competing interests. We declare we have no competing interest.

Funding. The authors wish to acknowledge present and past members of the Kabla and Charras laboratories for stimulating discussions. A.B. and A.K. wish to acknowledge Louis Kaplan for his important contribution to the development of the RHEOS package and Arran Fernandez for stimulating discussions on fractional calculus. A.B and J.F. were funded by BBSRC grant nos. (BB/M003280 and BB/M002578) to G.C. and A.K. N.K. was funded by the Rosetrees Trust, the UCL Graduate School, the EPSRC funded doctoral training programme CoMPLEX and the European Research Council (ERC-CoG MolCellTissMech, agreement no. 647186 to G.C.). N.K. was also in receipt of a UCL Overseas Research Scholarship. G.C. is supported by a consolidator grant from the European Research Council (MolCellTissMech, agreement no. 647186). The work was supported by BBSRC grant nos. (BB/K018175/1, BB/M003280 and BB/M002578).

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
