## [Reviewer comments · Royal Society Open Science]

Review History

RSOS-190920.R0 (Original submission)

Review form: Reviewer 1 (Roberto Cerbino)

Is the manuscript scientifically sound in its present form?

Yes

Are the interpretations and conclusions justified by the results?

Yes

Is the language acceptable?

Yes

Do you have any ethical concerns with this paper?

No

Have you any concerns about statistical analyses in this paper?

No

Recommendation?

Major revision is needed (please make suggestions in comments)

Comments to the Author(s)

The manuscript "A unified rheological model for cells and cellularised materials" by A. Bonfanti et al. describes a mixed rheological model combining traditional viscoelastic building blocks with a fractional element (spring-pot) to describe the rheological response of cell collectives.

The idea of using fractional calculus and the spring-pot in conjunction with more traditional viscoelastic elements is really interesting and enables one capturing a complex relaxation process with a reduced number of parameters. The price one has to pay for this simplification is two-fold: (i) the link of the model parameters with the physical properties of the system remains obscure; (ii) the computational complexity introduced with fractional calculus is non-negligible. In my view, the Authors do a very good job in trying to mitigate these two issues. First, they try to link the model parameters with the system properties by testing the model with a variety of different systems, in addition to the MDCK epithelial monolayer that was initially used to develop and validate the model. Also, they make publicly available a software package written in the Julia programming language that provides tools for analyzing rheological data according to the model proposed here.

The manuscript reads quite well and is well organized.

There are a few issues that I would like to raise, in no particular order:

1) Judging from Fig.1, the Standard Linear Solid model is bad not only for short times but also for long times. In essence, it seems to me that one can just use it to capture the time scale of the relaxation and not much more than that because it fails both for short and long times.

2) some claims are a bit overstretched and I think that toning them down a bit would do well to the overall impact of the paper. An example (not the only one I am afraid) can be found at page 6, line 58 where the agreement in Fig. 3a between experimental results and analytical prediction is described as excellent. I would not think of a single criterion according to which the above mentioned agreement could be defined as excellent. The agreement in Fig.3a is, at the very best, fair and I do not believe that being honest about it would diminish the manuscript value. I think that a critical reading may reveal other instances of such kind in the present version of the manuscript

3) the way that the results obtained with different treatments are currently presented is unsatisfactory. In the main text only Y27632 and DMSO are mentioned (and with a bit of confusion in my view), whereas in the SM the effect of other treatments (CK666 and SMIFH2) is studied. I believe that all the treatments should be mentioned in the main text by explaining clearly why they were used and what is the outcome of the experiments, by referring to the SM for plots and details that cannot fit into the main text. All this information is currently contained in the figure legends, which is not a good practice. Also, the only reference currently made in the main text to the different treatments reads "Other treatments have been applied to monolayers to examine the role of actin network organization and crosslinkers, without observing significant variations in the relaxation response of monolayers", which is a quite questionable statement. Y27632 treated cells, at the very least, exhibit substantially different k and c_{β} compared to the others. The Authors should revise this part of their story to improve the clarity and impact of their message.

4) the main limit of the model lies in the fact that it does not take into account explicitly the active contribution of the cells to the rheology. I find surprising that there is no mention of this fact in the whole manuscript and I urge the Authors to add a Section/Paragraph explicitly devoted to

treating this point, which is quite relevant in consideration of the fact that many other models do explicitly include active stresses in the cell collective rheology. In this respect, it would be extremely interesting to understand what can we learn from the fact that the same model describes both control cells and treated cells and what the Authors think is the effect of the system activity at the model-level.

In light of the previous comments, I consider this work of potentially high value and interest for the readership of RSOS but I recommend that the Authors address the above-outlined issues, which are currently preventing me from giving a positive recommendation for immediate publication.

Review form: Reviewer 2

Is the manuscript scientifically sound in its present form?

Yes

Are the interpretations and conclusions justified by the results?

Yes

Is the language acceptable?

Yes

Do you have any ethical concerns with this paper?

No

Have you any concerns about statistical analyses in this paper?

No

Recommendation?

Major revision is needed (please make suggestions in comments)

Comments to the Author(s)

GENERAL COMMENTS:

pros:

+ the overall discussion and presentation of data and model looks reasonable
 + a potentially interesting phenomenological model that overcomes many problems of certain other phenomenological models (consistency, clear mathematical formulation, reasonable simplicity, applicable to many different datasets) is introduced and its predictive power is demonstrated

suggestions:

* 4 parameters may not sound much for a phenomenological model but still provides a lot of freedom for (over-)fitting
 * limitations of the model should be discussed in greater detail

cons:

- the physical interpretation of the model is not discussed well enough to judge its potential beyond mere curve-fitting

- no comparison is made with more bottom-up models aspiring to provide a better founded unified approach to cell-rheological behaviour. A brief review by Eugene Terentjev and coworkers in *Soft Matter* and a longer review by Chase Broedersz and Fred MacKintosh in *Review of Modern Physics*, both from 2014, provide a good overview of these models.
- reasons for the universal description of so many data sets are not discussed

SUMMARY:

In summary, while the approach discussed in the present contribution may have its merits, the authors fail to provide a comparison with state-of-the-art modelling approaches from the literature. I think that the paper might become suitable for publication with such an in-depth discussion and comparison added by the authors.

MORE SPECIFIC COMMENTS:

"We further examined if the model could capture the relaxation response of epithelial sheets in which myosin contractility, one of the most important component controlling cellular mechanical properties, was inhibited [34] ..."

The discussion is a bit confusing: If I understood correctly, the treatment (at least partially) destroy the actin network (instead of only disabling myosin activity), thus drastic changes are expected. Please discuss this. A reduced value in "stiffness" k could also be related to the loss of a major cytoskeletal component.

Further, in the SI it is stated that the change in viscosity is not significant, please discuss this more.

Is there some redundancy in you model? Do other parameter combinations fit the data equally or nearly equally well?

What are other possible interpretations of this, then?

"We can observe that the material parameters are almost constant until roughly 30%"

Do you have an explanation for this large linear regime? Or is this typical for your system. Please discuss.

"The experimental results and the analytical predictions are in excellent agreement with no free parameters."

Could you please back up this statement by highlight the mean and median of the experimental data in fig.3a?

"Fig. 3 (b) The creep response of epithelial monolayers: fractional model response and experimental data"

The experimental data is not visible in (b).

"We could also capture the creep response of a single muscle cell exhibiting a two-power-law behaviour, and predict its storage and loss modulus (figure S14 (a)-(b))."

I cannot see a double powerlaw in your data (rather a single power-law). Further, I do not see how your model can account for double powerlaw behaviour. Please discuss.

Discussion of Fig. 5

The cursory discussion of the inelastic or plastic effects is weak. What do the authors want to say here? I reckon that some models discussed in the literature, such as the inelastic GWLC, would probably provide a physically more intuitive interpretation of the data. Please discuss.

Decision letter (RSOS-190920.R0)

08-Jul-2019

Dear Dr Kabla,

The editors assigned to your paper ("A unified rheological model for cells and cellularised materials") have now received comments from reviewers. We would like you to revise your paper in accordance with the referee and Associate Editor suggestions which can be found below (not including confidential reports to the Editor). Please note this decision does not guarantee eventual acceptance.

Please submit a copy of your revised paper before 31-Jul-2019. Please note that the revision deadline will expire at 00.00am on this date. If we do not hear from you within this time then it will be assumed that the paper has been withdrawn. In exceptional circumstances, extensions may be possible if agreed with the Editorial Office in advance. We do not allow multiple rounds of revision so we urge you to make every effort to fully address all of the comments at this stage. If deemed necessary by the Editors, your manuscript will be sent back to one or more of the original reviewers for assessment. If the original reviewers are not available, we may invite new reviewers.

- Data accessibility

It is a condition of publication that all supporting data are made available either as supplementary information or preferably in a suitable permanent repository. The data accessibility section should state where the article's supporting data can be accessed. This section

should also include details, where possible of where to access other relevant research materials such as statistical tools, protocols, software etc can be accessed. If the data have been deposited in an external repository this section should list the database, accession number and link to the DOI for all data from the article that have been made publicly available. Data sets that have been deposited in an external repository and have a DOI should also be appropriately cited in the manuscript and included in the reference list.

If you wish to submit your supporting data or code to Dryad (<http://datadryad.org/>), or modify your current submission to dryad, please use the following link:
<http://datadryad.org/submit?journalID=RSOS&manu=RSOS-190920>

- **Competing interests**

- **Authors' contributions**

- **Acknowledgements**

- **Funding statement**

Kind regards,

Alice Power

Editorial Coordinator

on behalf of Pietro Cicuta (Subject Editor)

Editor's comments (Dr Pietro Cicuta):

The two referees have given a detailed list of points to be addressed.

Comments to Author:

Reviewers' Comments to Author:

Reviewer: 1

Comments to the Author(s)

The manuscript "A unified rheological model for cells and cellularised materials" by A. Bonfanti et al. describes a mixed rheological model combining traditional viscoelastic building blocks with a fractional element (spring-pot) to describe the rheological response of cell collectives.

The idea of using fractional calculus and the spring-pot in conjunction with more traditional viscoelastic elements is really interesting and enables one capturing a complex relaxation process with a reduced number of parameters. The price one has to pay for this simplification is two-fold: (i) the link of the model parameters with the physical properties of the system remains obscure; (ii) the computational complexity introduced with fractional calculus is non-negligible. In my view, the Authors do a very good job in trying to mitigate these two issues. First, they try to link the model parameters with the system properties by testing the model with a variety of different systems, in addition to the MDCK epithelial monolayer that was initially used to develop and validate the model. Also, they make publicly available a software package written in the Julia programming language that provides tools for analyzing rheological data according to the model proposed here.

The manuscript reads quite well and is well organized.

There are a few issues that I would like to raise, in no particular order:

1) Judging from Fig.1, the Standard Linear Solid model is bad not only for short times but also for long times. In essence, it seems to me that one can just use it to capture the time scale of the relaxation and not much more than that because it fails both for short and long times.

2) some claims are a bit overstretched and I think that toning them down a bit would do well to the overall impact of the paper. An example (not the only one I am afraid) can be found at page 6, line 58 where the agreement in Fig. 3a between experimental results and analytical prediction is described as excellent. I would not think of a single criterion according to which the above mentioned agreement could be defined as excellent. The agreement in Fig.3a is, at the very best, fair and I do not believe that being honest about it would diminish the manuscript value. I think that a critical reading may reveal other instances of such kind in the present version of the manuscript

3) the way that the results obtained with different treatments are currently presented is unsatisfactory. In the main text only Y27632 and DMSO are mentioned (and with a bit of confusion in my view), whereas in the SM the effect of other treatments (CK666 and SMIFH2) is studied. I believe that all the treatments should be mentioned in the main text by explaining clearly why they were used and what is the outcome of the experiments, by referring to the SM for plots and details that cannot fit into the main text. All this information is currently contained in the figure legends, which is not a good practice. Also, the only reference currently made in the main text to the different treatments reads "Other treatments have been applied to monolayers to examine the role of actin network organization and crosslinkers, without observing significant variations in the relaxation response of monolayers", which is a quite questionable statement.

Y27632 treated cells, at the very least, exhibit substantially different k and c_{β} compared to the others. The Authors should revise this part of their story to improve the clarity and impact of their message.

4) the main limit of the model lies in the fact that it does not take into account explicitly the active contribution of the cells to the rheology. I find surprising that there is no mention of this fact in the whole manuscript and I urge the Authors to add a Section/Paragraph explicitly devoted to treating this point, which is quite relevant in consideration of the fact that many other models do explicitly include active stresses in the cell collective rheology. In this respect, it would be extremely interesting to understand what can we learn from the fact that the same model describes both control cells and treated cells and what the Authors think is the effect of the system activity at the model-level.

In light of the previous comments, I consider this work of potentially high value and interest for the readership of RSOS but I recommend that the Authors address the above-outlined issues, which are currently preventing me from giving a positive recommendation for immediate publication.

Reviewer: 2

Comments to the Author(s)

GENERAL COMMENTS:

pros:

- + the overall discussion and presentation of data and model looks reasonable
- + a potentially interesting phenomenological model that overcomes many problems of certain other phenomenological models (consistency, clear mathematical formulation, reasonable simplicity, applicable to many different datasets) is introduced and its predictive power is demonstrated

suggestions:

- * 4 parameters may not sound much for a phenomenological model but still provides a lot of freedom for (over-)fitting
- * limitations of the model should be discussed in greater detail

cons:

- the physical interpretation of the model is not discussed well enough to judge its potential beyond mere curve-fitting
- no comparison is made with more bottom-up models aspiring to provide a better founded unified approach to cell-rheological behaviour. A brief review by Eugene Terentjev and coworkers in *Soft Matter* and a longer review by Chase Broedersz and Fred MacKintosh in *Review of Modern Physics*, both from 2014, provide a good overview of these models.
- reasons for the universal description of so many data sets are not discussed

SUMMARY:

In summary, while the approach discussed in the present contribution may have its merits, the authors fail to provide a comparison with state-of-the-art modelling approaches from the literature. I think that the paper might become suitable for publication with such an in-depth discussion and comparison added by the authors.

MORE SPECIFIC COMMENTS:

"We further examined if the model could capture the relaxation response of epithelial sheets in which myosin contractility, one of the most important component controlling cellular mechanical properties, was inhibited [34] ..."

The discussion is a bit confusing: If I understood correctly, the treatment (at least partially) destroy the actin network (instead of only disabling myosin activity), thus drastic changes are expected. Please discuss this. A reduced value in "stiffness" k could also be related to the loss of a major cytoskeletal component.

Further, in the SI it is stated that the change in viscosity is not significant, please discuss this more.

Is there some redundancy in you model? Do other parameter combinations fit the data equally or nearly equally well?

What are other possible interpretations of this, then?

"We can observe that the material parameters are almost constant until roughly 30%"

Do you have an explanation for this large linear regime? Or is this typical for your system. Please discuss.

"The experimental results and the analytical predictions are in excellent agreement with no free parameters."

Could you please back up this statement by highlight the mean and median of the experimental data in fig.3a?

"Fig. 3 (b) The creep response of epithelial monolayers: fractional model response and experimental data"

The experimental data is not visible in (b).

"We could also capture the creep response of a single muscle cell exhibiting a two-power-law behaviour, and predict its storage and loss modulus (figure S14 (a)-(b))."

I cannot see a double powerlaw in your data (rather a single power-law). Further, I do not see how your model can account for double powerlaw behaviour. Please discuss.

Discussion of Fig. 5

The cursory discussion of the inelastic or plastic effects is weak. What do the authors want to say here? I reckon that some models discussed in the literature, such as the inelastic GWLC, would probably provide a physically more intuitive interpretation of the data. Please discuss.

Author's Response to Decision Letter for (RSOS-190920.R0)

See Appendix A.

RSOS-190920.R1 (Revision)

Review form: Reviewer 1 (Roberto Cerbino)

Is the manuscript scientifically sound in its present form?

Yes

Are the interpretations and conclusions justified by the results?

Yes

Is the language acceptable?

Yes

Do you have any ethical concerns with this paper?

No

Have you any concerns about statistical analyses in this paper?

No

Recommendation?

Accept as is

Comments to the Author(s)

The revised version of the manuscript addresses all the important issues raised by the reviewers during the previous round. The resulting description is now clearer and likely more impactful. I thus happily recommend publication of this interesting contribution in RSOS.

Review form: Reviewer 2

Is the manuscript scientifically sound in its present form?

Yes

Are the interpretations and conclusions justified by the results?

Yes

Is the language acceptable?

Yes

Do you have any ethical concerns with this paper?

No

Have you any concerns about statistical analyses in this paper?

No

Recommendation?

Accept with minor revision (please list in comments)

Comments to the Author(s)

The authors have made an effort to improve their manuscript, and I would no longer object against publication, although I am still not impressed by how they relate their model to other models from the literature. I think that this remains one of the main weaknesses of the paper. Finally, the statement "The fact that the same model could capture the behaviour of a wide range of biological systems, some actively controlling their behaviour (single cells or tissues) and others passive (single collagen fibrils), demonstrates that there is not a unique biophysical component responsible for the power-law response, and reinforces the need for a holistic approach to understand power law rheology." sounds mystifying and vague (if not nonsensical) to me. Please either provide a concrete and technically explicit discussion of what you precisely mean or suppress this statement.

Decision letter (RSOS-190920.R1)

16-Oct-2019

Dear Dr Kabla:

On behalf of the Editors, I am pleased to inform you that your Manuscript RSOS-190920.R1 entitled "A unified rheological model for cells and cellularised materials" has been accepted for publication in Royal Society Open Science subject to minor revision in accordance with the referee suggestions. Please find the referees' comments at the end of this email.

The reviewers and Subject Editor have recommended publication, but also suggest some minor revisions to your manuscript. Therefore, I invite you to respond to the comments and revise your manuscript.

- Ethics statement

- Data accessibility

If you wish to submit your supporting data or code to Dryad (<http://datadryad.org/>), or modify your current submission to dryad, please use the following link:
<http://datadryad.org/submit?journalID=RSOS&manu=RSOS-190920.R1>

- Competing interests

- Authors' contributions

- Acknowledgements

- Funding statement

Because the schedule for publication is very tight, it is a condition of publication that you submit the revised version of your manuscript before 25-Oct-2019. Please note that the revision deadline will expire at 00.00am on this date. If you do not think you will be able to meet this date please let me know immediately.

- 1) A text file of the manuscript (tex, txt, rtf, docx or doc), references, tables (including captions) and figure captions. Do not upload a PDF as your "Main Document".
- 2) A separate electronic file of each figure (EPS or print-quality PDF preferred (either format should be produced directly from original creation package), or original software format)

3) Included a 100 word media summary of your paper when requested at submission. Please ensure you have entered correct contact details (email, institution and telephone) in your user account

4) Included the raw data to support the claims made in your paper. You can either include your data as electronic supplementary material or upload to a repository and include the relevant doi within your manuscript

5) All supplementary materials accompanying an accepted article will be treated as in their final form. Note that the Royal Society will neither edit nor typeset supplementary material and it will be hosted as provided. Please ensure that the supplementary material includes the paper details where possible (authors, article title, journal name).

Kind regards,
Anita Kristiansen
Editorial Coordinator
Royal Society Open Science
openscience@royalsociety.org

on behalf of Pietro Cicuta (Subject Editor)
openscience@royalsociety.org

Associate Editor Comments to Author (Dr Pietro Cicuta):

Associate Editor:

Comments to the Author:

Both reviewers have commented on the improvement of the manuscript. There is a small suggestion from one of the reviewers which I think should be addressed, and also a question as to whether the link to the online data is active.

Reviewer comments to Author:

Reviewer: 1

Comments to the Author(s)

The revised version of the manuscript addresses all the important issues raised by the reviewers during the previous round. The resulting description is now clearer and likely more impactful. I thus happily recommend publication of this interesting contribution in RSOS.

Reviewer: 2

Comments to the Author(s)

The authors have made an effort to improve their manuscript, and I would no longer object against publication, although I am still not impressed by how they relate their model to other

models from the literature. I think that this remains one of the main weaknesses of the paper. Finally, the statement "The fact that the same model could capture the behaviour of a wide range of biological systems, some actively controlling their behaviour (single cells or tissues) and others passive (single collagen fibrils), demonstrates that there is not a unique biophysical component responsible for the power-law response, and reinforces the need for a holistic approach to understand power law rheology." sounds mystifying and vague (if not nonsensical) to me. Please either provide a concrete and technically explicit discussion of what you precisely mean or suppress this statement.

Author's Response to Decision Letter for (RSOS-190920.R1)

See Appendix B.

Decision letter (RSOS-190920.R2)

22-Nov-2019

Dear Dr Kabla,

It is a pleasure to accept your manuscript entitled "A unified rheological model for cells and cellularised materials" in its current form for publication in Royal Society Open Science. The comments of the reviewer(s) who reviewed your manuscript are included at the foot of this letter.

Kind regards,
Lianne Parkhouse
Editorial Coordinator
Royal Society Open Science
openscience@royalsociety.org

on behalf of Dr Pietro Cicuta (Associate Editor) and Pietro Cicuta (Subject Editor)
openscience@royalsociety.org

Associate Editor Comments to Author (Dr Pietro Cicuta):

Congratulations - paper ready for acceptance.

Appendix A

Dear Editor,

Many thanks for giving us the opportunity to revise our manuscript. We are very grateful to the referees for their comments that allowed us to make significant improvements to the manuscript.

Following the recommendations from the referees, we rewrote most of the discussion in order to better describe the scope and impact of the paper, and clarify how it relates to the biophysical interpretation of the data as well as other models. We also better explain how the model parameters relate to corresponding features of the experimental rheological response of monolayers (new figure 1), therefore justifying that we are not overfitting. The effect of pharmacological treatment is also better integrated with the paper. We highlighted where appropriate the limitations of the model. A large number of minor changes were included to account for more straight-forward comments. Specific responses to all reviewers' comments are included with the resubmission.

We hope very much that you will find that these changes adequately address the referees' comments and that the revised paper will be suitable for publication.

We look forward to hearing back from you. Please let me know if you have any further questions or queries about this work.

Best wishes,

Alexandre Kabla, on the behalf of all co-authors.

Response to the referees' comments

The authors would like to thank all the reviewers for their comments on the manuscript. We provide below responses in blue ink to all suggestions and criticisms raised by the referees.

Reviewers' Comments to Author:

Reviewer: 1

Comments to the Author(s)

The manuscript "A unified rheological model for cells and cellularised materials" by A. Bonfanti et al. describes a mixed rheological model combining traditional viscoelastic building blocks with a fractional element (spring-pot) to describe the rheological response of cell collectives.

The idea of using fractional calculus and the spring-pot in conjunction with more traditional viscoelastic elements is really interesting and enables one capturing a complex relaxation process with a reduced number of parameters. The price one has to pay for this simplification is two-fold: (i) the link of the model parameters with the physical properties of the system remains obscure; (ii) the computational complexity introduced with fractional calculus is non-negligible. In my view, the Authors do a very good job in trying to mitigate these two issues. First, they try to link the model parameters with the system properties by testing the model with a variety of different systems, in addition to the MDCK epithelial monolayer that was initially used to develop and validate the model. Also, they make publicly available a software package written in the Julia programming language that provides tools for analyzing rheological data according to the model proposed here. The manuscript reads quite well and is well organized.

There are a few issues that I would like to raise, in no particular order:

1) Judging from Fig.1, the Standard Linear Solid model is bad not only for short times but also for long times. In essence, it seems to me that one can just use it to capture the time scale of the relaxation and not much more than that because it fails both for short and long times.

The referee is right that the main point of this figure was simply to illustrate the presence of a time-scale, which can be captured by the SLS model. In response to this comment and another comment from Reviewer 2, this figure has been substituted with an alternative representation of the data which clearly shows the relevant regimes. The comparison between the SLS fit and our proposed model remains available in Supplementary Information (Fig S5).

2) some claims are a bit overstretched and I think that toning them down a bit would do well to the overall impact of the paper. An example (not the only one I am afraid) can be found at page 6, line 58 where the agreement in Fig. 3a between experimental results and analytical prediction is described as excellent. I would not think of a single criterion according to which the above mentioned agreement could be defined as excellent. The agreement in Fig.3a is, at the very best, fair and I do not believe that being honest about it would diminish the manuscript value. I think that a critical reading may reveal other instances of such kind in the present version of the manuscript

We agree with the reviewer. The manuscript has been corrected accordingly. Here are a few examples of changes implemented.

(Section 3, 1st paragraph): “The experimental results and the model predictions are in agreement with no free parameters.”

(Section 4, 2nd paragraph): “we calculated the analytical expression of the complex modulus associated with our model and fitted the experimental data again with a good agreement”

(Conclusions, 1st paragraph): “The model, calibrated against experimental relaxation data of suspended MDCK monolayers, could predict the monolayer's response to other mechanical tests with good accuracy.”

3) the way that the results obtained with different treatments are currently presented is unsatisfactory. In the main text only Y27632 and DMSO are mentioned (and with a bit of confusion in my view), whereas in the SM the effect of other treatments (CK666 and SMIFH2) is studied. I believe that all the treatments should be mentioned in the main text by explaining clearly why they were used and what is the outcome of the experiments, by referring to the SM for plots and details that cannot fit into the main text. All this information is currently contained in the figure legends, which is not a good practice. Also, the only reference currently made in the main text to the different treatments reads “Other treatments have been applied to monolayers to examine the role of actin network organization and crosslinkers, without observing significant variations in the relaxation response of monolayers”, which is a quite questionable statement. Y27632 treated cells, at the very least, exhibit substantially different k and c_{β} compared to the others. The Authors should revise this part of their story to improve the clarity and impact of their message.

We thank the reviewer for this comment. We clarified the paper as a result. Y27632 is the treatment that impacted significantly the rheological response (which is why this data was already included in the main paper). We have now listed in the main paper the other treatments tested, with a succinct statement about what they affect. The corresponding data remains in supplementary material only. We added a section in supplementary material (section 4) to provide more context in addition to the data. We would like to emphasise here that this primary data has been published already. Only the analysis is novel, and is in broad agreement with an earlier analysis. So we do not want to exaggerate the importance of this data in the paper.

4) the main limit of the model lies in the fact that it does not take into account explicitly the active contribution of the cells to the rheology. I find surprising that there is no mention of this fact in the whole manuscript and I urge the Authors to add a Section/Paragraph explicitly devoted to treating this point, which is quite relevant in consideration of the fact that many other models do explicitly include active stresses in the cell collective rheology. In this respect, it would be extremely interesting to understand what can we learn from the fact that the same model describes both control cells and treated cells and what the Authors think is the effect of the system activity at the model-level.

The referee is correct that the model does not explicitly capture active contributions, although many active cellular processes would contribute to setting the values of the measured parameters. However, the paper is not primarily concerned with the validation of specific physical assumptions. A broad range of materials exhibit power law responses consistent with the model, including single collagen fibrils, as indicated by fig 4. Hence, the power law behaviour cannot be specifically related

to particular active or passive behaviour, but seems to be a trait that can arise from a number of biological constructs. It is also common in soft materials in general. We believe it is the strength of our method to provide a parameterization of data that does not depend on underlying assumptions, but nevertheless enables comparison across systems. This is why the issue of active behaviour is not prevalent in the paper. We also often do not have the means to be more specific with mechanisms at this stage.

The referee is nonetheless raising a fair point that needs clarification in the paper.

- We have added a sentence in the discussion to address specifically the case of prestress in monolayers and its link to the long time-scale elastic modulus of the material, drawing connections with recent work from our group (Wyatt, Fouchard et al., to appear in Nature Materials).
- We highlight that cortical contractility, controlling k , is an ATP driven active process (discussion).
- We also clarified in the conclusion that many processes, active or passive, could contribute to the emerging rheological behaviour.

In light of the previous comments, I consider this work of potentially high value and interest for the readership of RSOS but I recommend that the Authors address the above-outlined issues, which are currently preventing me from giving a positive recommendation for immediate publication.

We thank the reviewer for this supportive statement and hope to have addressed their concerns.

Reviewer: 2

Comments to the Author(s)

GENERAL COMMENTS:

pros:

+ the overall discussion and presentation of data and model looks reasonable
+ a potentially interesting phenomenological model that overcomes many problems of certain other phenomenological models (consistency, clear mathematical formulation, reasonable simplicity, applicable to many different datasets) is introduced and its predictive power is demonstrated

suggestions:

* 4 parameters may not sound much for a phenomenological model but still provides a lot of freedom for (over-)fitting

We are confident the model does not overfit. To clarify this, Figure 1 has been replaced by a more explicit representation of these regimes. The text has been changed accordingly.

To be more specific, from fig 1, 5 parameters can be a priori extracted from the three red lines: on panels (a), we can extract an elastic constant k , on panel (b) we have an amplitude B and the characteristic time τ of the exponential regime, $B e^{-t/\tau}$, and on panel (c) the amplitude A and exponent β of a power law, $A t^\beta$. These five parameters are a priori independent. However, our model only has four parameters, which indicates that these parameters are not independent in

practice; the exponential regime only acts as cut-off process, whose magnitude B is related to A . The four parameters of the model therefore correspond to: (i) the level of the final plateau, (ii) the time-scale beyond which the relaxation function becomes negligible (exponential cut-off time), (iii) the power-law exponent at short time scales, and (iv) the overall magnitude of the dissipative branch. All these parameters can be independently extracted from the data.

* limitations of the model should be discussed in greater detail

We thank the reviewer for his comments. The main limitation of the model is the assumption of linearity. We had already a supplementary figure testing this hypothesis in the MDCK monolayer, but this limitation is now mentioned in the conclusion. We also included a statement about the fact that active stresses and active processes in general are not explicitly accounted for in the model. Another limitation is of course that the model is phenomenological, a point further discussed below.

cons:

- the physical interpretation of the model is not discussed well enough to judge its potential beyond mere curve-fitting

- no comparison is made with more bottom-up models aspiring to provide a better founded unified approach to cell-rheological behaviour. A brief review by Eugene Terentjev and coworkers in *Soft Matter* and a longer review by Chase Broedersz and Fred MacKintosh in *Review of Modern Physics*, both from 2014, provide a good overview of these models.

- reasons for the universal description of so many data sets are not discussed

We thank the referee for their thoughtful comments about the scope of our work.

The referee is right that the physical interpretation of the data is at this stage limited.

The aim of this manuscript is to introduce a viscoelastic model that allows a more systematic and reproducible analysis of experimental data. Providing an interpretation of the rheology in terms of microstructural processes would go beyond the scope of this paper. The work however goes beyond mere curve-fitting, in the sense that we demonstrate the experimental validity of a linear system's approach by extracting parameters and making predictions. This allows us to claim that parameters extracted from the fit are inherent properties of the material, even though we do not know their precise meaning.

We have nonetheless rewritten most of the discussion and conclusion to clarify this position, and better highlight how it relates to the biophysical interpretation of the data where we can. We discuss more in depth the role of cortical contractility on stiffness. The penultimate paragraph in section 4 addresses specifically the relationship with physical models.

As stated by the referee, the model has been successfully applied to capture the behaviour of components, both active and passive, at different scales (e.g. epithelial monolayers, single cells, collagen fibrils, cell cortex). This wide applicability may suggest that there is not a single physical interpretation for the origin of these universal behaviour. A statement in the conclusion makes this point more explicit.

SUMMARY:

In summary, while the approach discussed in the present contribution may have its merits, the authors fail to provide a comparison with state-of-the-art modelling approaches from the literature. I think that the paper might become suitable for publication with such an in-depth discussion and comparison added by the authors.

MORE SPECIFIC COMMENTS:

"We further examined if the model could capture the relaxation response of epithelial sheets in which myosin contractility, one of the most important component controlling cellular mechanical properties, was inhibited [34] ..."

The discussion is a bit confusing: If I understood correctly, the treatment (at least partially) destroy the actin network (instead of only disabling myosin activity), thus drastic changes are expected. Please discuss this. A reduced value in "stiffness" k could also be related to the loss of a major cytoskeletal component. Further, in the SI it is stated that the change in viscosity is not significant, please discuss this more. Is there some redundancy in you model? Do other parameter combinations fit the data equally or nearly equally well? What are other possible interpretations of this, then?

Y27632 primarily affects contractility (as ROCK inhibitor). We included data in supplementary material corresponding to treatments where actin polymerisation is directly affected, but these did not exhibit significant changes in the rheology. The viscosity changes with Y27632 are visible based on the mean or median values, but variations are statistically non-significant.

In response to this comment and a more specific request from reviewer 1, we clarified in the main paper the purpose of the different treatments, and added a section in supplementary material.

As discussed earlier, we do not believe that our model could be over-fitting the curves, especially for relaxation experiments where the power-law cut-off is visible (because the experiments were long enough to capture the transition between the two regimes).

"We can observe that the material parameters are almost constant until roughly 30%"
Do you have an explanation for this large linear regime? Or is this typical for your system. Please discuss.

We thank the reviewer for this important question. We included our own data on monolayers in the paper. We do not know how generic this threshold is. Understanding of the nonlinear regime of such a system is currently under investigation. Given previous results that confirmed the role of intermediate filaments at large deformation [1], we can speculate that they are involved in the nonlinear behaviour, but there may be other players..

[1] Harris, Andrew R., et al. "Characterizing the mechanics of cultured cell monolayers." *Proceedings of the National Academy of Sciences* 109.41 (2012): 16449-16454.

"The experimental results and the analytical predictions are in excellent agreement with no free parameters." Could you please back up this statement by highlight the mean and median of the experimental data in fig.3a?

Both referees pointed the subjective nature of this statement. The values are consistent both in trend and order of magnitude. There are considerable variations in stiffness from one monolayer to the next, which is why we did not average curves and displayed all the data available. It is fair to say that the agreement does not look exceptional on the figure. We initially had low expectations regarding the validity of this model, hence our over-enthusiastic description of the result. As rightly requested by referee 1, we toned down such statements in a few places.

"Fig. 3 (b) The creep response of epithelial monolayers: fractional model response and experimental data"

The experimental data is not visible in (b).

We apologise for the wrong caption. This has now been corrected.

"We could also capture the creep response of a single muscle cell exhibiting a two-power-law behaviour, and predict its storage and loss modulus (figure S14 (a)-(b))."

I cannot see a double powerlaw in your data (rather a single power-law). Further, I do not see how your model can account for double powerlaw behaviour. Please discuss.

We agree with the reviewer that this statement is confusing. The single muscle cell response shows an initial power-law response followed by an increase of slope. This two-phase behaviour is captured by the model via an initial response dominated by the spring-pot, followed by another regime dominated by the dashpot. In cases where the long time-scale elastic behaviour is negligible (k approximately zero), the creep response would exhibit two power laws (linear at long time scale). This has been clarified accordingly in the text.

Discussion of Fig. 5

The cursory discussion of the inelastic or plastic effects is weak. What do the authors want to say here? I reckon that some models discussed in the literature, such as the inelastic GWLC, would probably provide a physically more intuitive interpretation of the data. Please discuss.

We rewrote this section of the paper to better flag our message: having at our disposal a broader set of visco-elastic models influences the way we interpret experimental data. This is why we provided an alternative way to summarise some of the data presented in Bonakdar et al. We do not attempt to explain their results but demonstrate in a practical way how to adapt our model to account for the behaviour observed. We find it however remarkable that this data set appears to be qualitatively similar to single cell data obtained on MDCK or Hela cells. Being able to draw such comparison is a key outcome of our work, and hope that the revised manuscript makes this clearer.

Appendix B

Dear Editor,

many thanks for accepting the paper. Following the last round of reviews, we decided to simply remove the sentence that referee #2 found unclear, as suggested by them.

We added to the reference the web link to the data provided by Driad, hopefully it will be activated in due course. Let us know if more is needed.

Many thanks.

Alexandre Kabla.